# Evidence from fMRI Supports a Two-Phase Abstraction Process in Language Models

**Richard J. Antonello**[*]
Columbia University
New York City, New York
rja2163@columbia.edu

**Emily Cheng**[*]
Universitat Pompeu Fabra
Barcelona, Spain
emilyshana.cheng@upf.edu

## Abstract

Research has repeatedly demonstrated that intermediate hidden states extracted from large language models predict measured brain response to natural language stimuli. Yet, very little is known about the representation properties that enable this high prediction performance. Why is it the intermediate layers, and not the output layers, that are most capable for this unique and highly general transfer task? In this work, we show that evidence from language encoding models in fMRI supports the existence of a two-phase abstraction process within LLMs. We use geometric methods to show that this abstraction process naturally arises over the course of training a language model and that the first "composition" phase of this abstraction process is compressed into fewer layers as training continues. Finally, we demonstrate a strong correspondence between layerwise encoding performance and the intrinsic dimensionality of representations from LLMs. We give initial evidence that this correspondence primarily derives from the inherent compositionality of LLMs and not their next-word prediction properties.

## 1 Introduction

How do brains and machines take low-level information, such as a collection of sounds or words, and compose it into the rich tapestry of ideas and concepts that can be expressed in natural language? This question of composition, or abstraction, is at the heart of most studies of human language comprehension. Recent work has shown that representations from large language models (LLMs) are able to successfully model human brain activity at varying spatial and temporal resolutions with only a linear transformation [1, 2, 3, 4, 5, 6, 7]. This has led to questions about the reason for this brain-model similarity. Do LLMs and brains possess similar representations because they have similar learning properties or objectives? [8, 9, 1] Or is the similarity merely a consequence of shared abstraction, the ability to represent features not derivable from the lexical properties of language alone? [10]

In this work, we present new evidence that it is the abstractive, compositional properties of LLMs that drive predictivity between LLMs and brains. We do this by examining an underexplored and unexplained phenomenon of the similarity - the tendency for intermediate hidden layers of LLMs to be optimal for this linear transfer task. We show that an LLM layer's performance at predicting brain activity is strongly related to intrinsic dimensionality of that layer relative to other layers in the same network. Furthermore, we demonstrate that this relationship is itself an indicator that pretrained LLMs naturally split into an early *abstraction*, or composition, phase, and a later *prediction*, or extraction, phase, a result independently suggested in the LM interpretability literature [11, 12]. We suggest that it is the first abstraction phase, rather than the latter prediction phase, that primarily drives the observed correspondence between brains and LLMs.

---

[*]These authors contributed equally to this work.

Preprint.

Table 1: The average voxelwise product-moment correlations between representational dimensionality and encoding performance are shown for $I_d$, PCA-$d$ (variance threshold of 0.99), and PR-$d$. Across models, the correlation is generally high no matter the dimensionality measure. All values, except those marked with (*), are significant to $p < 10^{-3}$, as computed by a permutation test.

| | OPT-125M | OPT-1.3B | OPT-13B | Pythia-6.9B |
|---|---|---|---|---|
| GRIDE $I_d$ | 0.91 | **0.96** | 0.85 | **0.90** |
| PCA $d$ | 0.91 | 0.93 | **0.96** | 0.86 |
| PR $d$ | **0.94** | 0.82 | 0.85 | $-0.05^*$ |

## 2 Methods

We test the hypothesis that feature abstraction, not next-token prediction *per se*, drives brain-model similarity. To do so requires three observables. First, we measure the dependent variable, **(1)** brain-model representational similarity, by scoring the prediction performance of a learned linear mapping from LLM representations to brain activity. Then, we compute the **(2)** dimensionality of representations to measure abstract feature complexity over the LLM's layers. Finally, to test the alternate hypothesis that next-token prediction drives brain-LM similarity, as has been suggested by others [13, 8, 1], we compute the **(3)** *surprisal*, or next-token prediction error, from each layer.

### 2.1 Brain-model similarity

**fMRI data** We used publicly available functional magnetic resonance imaging (fMRI) data collected from 3 human subjects as they listened to 20 hours of English language podcast stories over Sensimetrics S14 headphones. Stories came from podcasts such as *The Moth Radio Hour*, *Modern Love*, and *The Anthropocene Reviewed*. Each 10-15 minute story was played during a separate scan. Subjects were not asked to make any responses, but simply to listen attentively to the stories. For encoding model training, each subject listened to roughly 95 different stories, giving 20 hours of data across 20 scanning sessions, or a total of ~33,000 datapoints for each voxel across the whole brain. Additional details of the MRI methods are summarized in Appendix D.

**Neural encoding model training** To train encoding models, we use the method described in [4]. High-level details of the method are summarized here. For each word in the stimulus set, activations were extracted by feeding that word and its immediate preceding context into the LLM. A sliding window was used to ensure each word received a minimum of 256 tokens of context. Activations were then downsampled using a Lanczos filter and FIR delays of 1,2,3 and 4 TRs were added to account for the hemodynamic lag in the BOLD signal. A linear projection from the downsampled, time-delayed features was trained using ridge regression. Encoding models were built using the OPT language model [14] (three sizes - 125M, 1.3B, 13B) and the 6.9B parameter deduped Pythia language model [15]. To study model training, 9 different Pythia model checkpoints were used (at 1K, 2K, 3K, 4K, 8K, 16K, 32K, 64K, and 143K training steps). Model details are summarized in Table E.1.

### 2.2 Dimensionality of neural manifolds

To measure representational complexity, we compute the *intrinsic dimensionality* $I_d$ as well as the linear *effective dimensionality* $d$ of activations at each layer. $I_d$ and $d$ describe different geometric properties of the representations: while the former is the dimension of the representations' underlying (nonlinear) manifold, the latter describes the number of linear directions that explain their variance up to a threshold. We will use *dimensionality* to refer to both $I_d$ and $d$, specifying when necessary.

We are interested in an LLM's behavior on a representative sample of natural language, so that the computed dimensionality is informative about the model's general linguistic processing. For all models, we compute the ID on $N = 10000$ 20-word contexts randomly sampled from Pythia's training data,[1] The Pile [16], over 5 random data samples. The model's tokenization scheme can produce sequences of variable length, so we aggregated representations at each layer by taking the last

---

[1]The training data for OPT are not publicly downloadable.

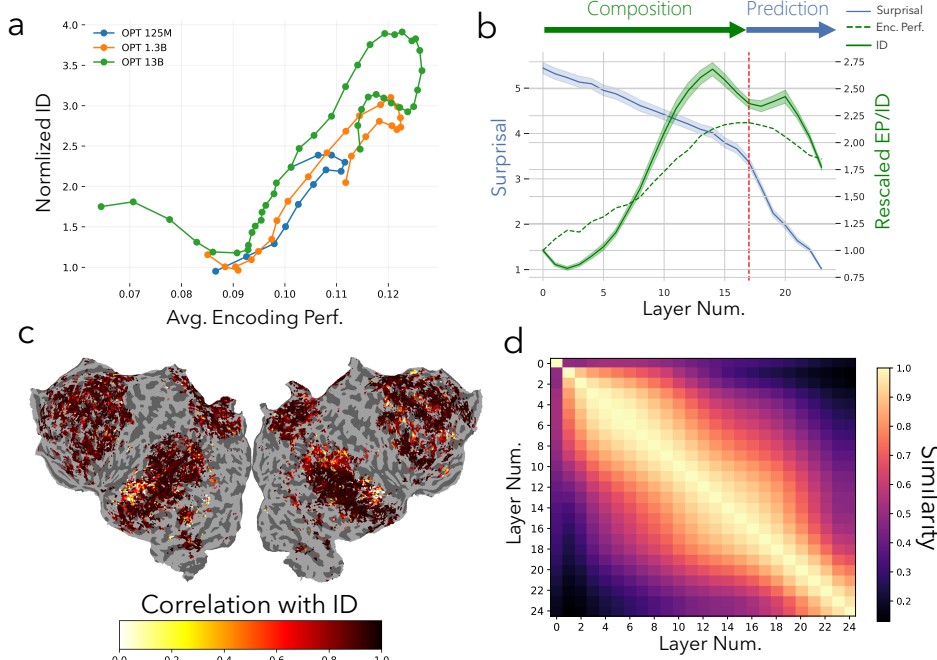

Figure 1: *Analyzing Layerwise Representational Trends*: **(a)** $I_d$ is well correlated with encoding performance across model sizes. $I_d$ is normalized here by the log of embedding size to account for power law scaling. **(b)** The abstract-predict phase transition at layer 17 is shown for OPT-1.3B. At the peak of encoding performance (red dashed line), the next-token prediction loss (blue curve) sharply decreases, corresponding with a decrease in encoding performance. **(c)** A flatmap of the brain, for one subject, is shown colored voxelwise by the correlation over layers between $I_d$ and encoding performance. With the exception of auditory cortex (bright), which captures low-level spectral information, encoding performance in brain regions thought to perform higher-level linguistic processing (dark) is well-captured by representational $I_d$.**(d)** The layer-wise representational similarity computed with linear CKA is shown for OPT-1.3B.

token representation in the model's *residual stream* [17]; this yields one $N \times D$ matrix of activations per layer, $D$ being the model's hidden dimension, or extrinsic dimension.

**Nonlinear ID estimation**    To compute $I_d$, we apply the Generalized Ratios Intrinsic Dimension Estimator (GRIDE) [18], an extension of the popular TwoNN estimator [19] to general scales. GRIDE operates on ratios $\mu_{i,2k,k} := r_{i,2k}/r_{i,k}$, where $r_{i,j}$ is the Euclidean distance between point $i$ and its $j^{th}$ neighbor. Assuming local uniform density up to the $2k^{th}$ neighbor, the ratios $\mu_{i,2k,k}$ follow a generalized Pareto distribution $f_{\mu_i,2k,k}(\mu) = \frac{I_d(\mu^{I_d}-1)^{k-1}}{B(k,k)\mu^{I_d(2k-1)+1}}$, where $B(\cdot, \cdot)$ is the beta function. The $I_d$ is then recovered by maximizing this likelihood over points $i$ for several candidate scales $k$.

Finally, in order to choose the proper $I_d$, a scale analysis over $k$, which controls the neighborhood size, is necessary: if $k$ is too small, the $I_d$ likely describes local noise, and if $k$ is too large, the curvature of the manifold will produce a faulty estimate. Instead, it is recommended to choose a $k$ for which the $I_d$ is stable [18]. We provide an example of such a scale analysis in Appendix B.

**Linear dimensionality estimation**    In addition to nonlinear $I_d$, we computed linear effective dimensionality $d$ two ways: using PCA with variance cutoff 0.99 [20], and the Participation Ratio (PR), defined as $(\sum_i \lambda_i)^2/(\sum_i \lambda_i^2)$ [21]. By definition, $I_d$-dimensional manifolds can be embedded in $d \geq I_d$ dimensions, so we expect that $d \geq I_d$.

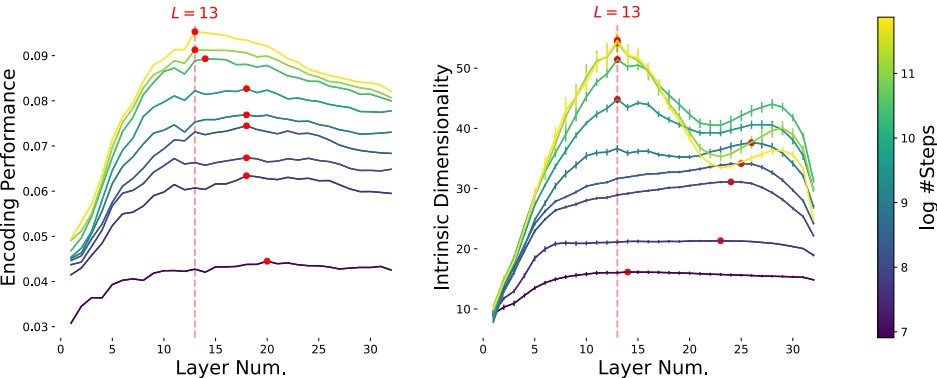

Figure 2: *Encoding Performance and Intrinsic Dimensionality Peaks Manifest Concurrently over Training*: **(a)** - The evolution of layerwise encoding performance over training of the Pythia 6.9B model is shown. A peak is reached at layer 13 of the model. **(b)** -Likewise, a peak in $I_d$ at layer 13 manifests over training. Red dots in each figure denote maximal layers for the respective metric.

## 2.3 Measuring layerwise surprisal

To determine whether predictive coding explains brain-LLM representational similarity, on The Pile, we computed the next token's surprisal from intermediate layers using TunedLens [22]. TunedLens learns an affine mapping from an intermediate layer to the vocabulary space in order to predict the next token, indicating how much intermediate layers (linearly) represent next-token identity. See Appendix C for implementation details.

## 3 Results

Layerwise encoding performance and representational dimensionality, linear and nonlinear, are highly correlated across brain areas involved in linguistic processing. Table 1 shows the correlation between encoding performance and dimensionality averaged over all voxels, and Figure 1a shows the correlation between average encoding performance and (normalized) $I_d$ for OPT models. The positive relationship, $\rho = 0.85$, between $I_d$ and encoding performance suggests that in trained models, the $I_d$ of layer activations captures linguistic feature complexity needed to support language comprehension.

Figure 1b overlays, for OPT-1.3B, the encoding performance, $I_d$, and next-token prediction loss computed from each layer. Encoding performance peaks at layer 17, which exactly marks a sharp downwards turn in prediction loss. While Cheng et al. [12] showed pre-$I_d$-peak layers to extract syntactic and semantic features, our results also suggest a functional shift post-$I_d$-peak to next-token prediction. The sharp transition from abstraction to prediction is observed across OPT model sizes, but it is more gradual for Pythia (see Appendix F.1). To further evidence a transition in layer function, we report inter-layer representational similarity via linear Centered Kernel Alignment [23]. In Figure 1d, where lighter is more similar, the $I_d$ peak approximately marks a point where preceding layers are no longer similar to following ones. Results hold across models, see Appendix F.3.

Figure 1c shows, for one subject, the voxelwise correlation of $I_d$ with encoding performance across layers (dark red is better). Except for the primary auditory cortex, which processes low-level auditory information, encoding performance in brain areas thought to handle higher-level linguistic processing is well-predicted by $I_d$. Results hold across subjects and models, see Appendix F.2.

The relationship between encoding performance and $I_d$ arises nontrivially from learning. Figure 2 plots the encoding performance (left) and $I_d$ (right) across layers over the course of training for Pythia-6.9B (each curve is a different checkpoint). We confirm an existing result from the literature that the $I_d$ peak emerges and that $I_d$ generally grows for all layers over training (Figure 2 right) [12]. Furthermore, encoding performance and $I_d$ increase at similar rates over training, seen by similar positions of the checkpoint curves in the two plots. The two plots are globally correlated with $\rho = 0.94$. Lastly, the location of the $I_d$ peak (red dots, right), changes over training, eventually settling at the same layers for peak encoding performance (red dots, left). This rules out that the $I_d$ peak trivially reflects the Transformer architecture, e.g., layer index.

## 4 Discussion

Recent studies of the properties of language encoding models have observed that the intermediate layers of LLMs, rather than the output layers, have the highest linear similarity to measured brain activity. This is true regardless of the scanning modality (be it fMRI [4], ECoG [24], or MEG [8]), and regardless of the chosen LLM. Despite this very frequently observed trend, little research has been dedicated to explaining this phenomenon. Yet, an understanding of this trend would greatly benefit our understanding of both brains and LLMs, not least because layerwise differences in LLMs have highly useful epistemic properties. LLM layers are invariant to many confounding variables - each layer has seen the same data in the same order, has an identical architecture, was trained on the same loss term, and built using the same hyperparameters. Therefore, differences between layers can only arise either as a result of the compositional nature of the transition from earlier layers to later ones, or due to the "time pressure" exerted by the loss term on the final output layers.

These competing pressures, to first build up the most comprehensive representation of the input text possible, and to then ultimately use this representation to resolve towards a distribution over predicted next word outputs, have opposite effects, as we demonstrate here. The composition effect leads to a increase in encoding performance and dimensionality, whereas the prediction effect narrows the dimensionality to the detriment of encoding. Furthermore, we observe that as models get larger and more thoroughly trained, the best layer for encoding slowly drifts to earlier in the model, perhaps suggesting a saturation effect for this initial compositional phase.

What conclusions should we draw from this? Firstly, that it is not likely to be the autoregressive nature of language models that drives brain-model similarity [9, 1, 10]. As models get more potent at prediction, their most predictive and most descriptive layers drift apart.[2] Secondly, we can draw that the multi-phase abstraction process in LLMs that has been proposed independently by other authors [12, 11] is supported by evidence from the only other system known to effectively reason with complex language, the human brain. As the present work only tests two model families, it will be necessary to test more models for conclusions to hold in the general case.

From a practical perspective, conclusions point to a potential new avenue for improving the performance of encoding models. If the spectral properties of different LLM layers can be measured and efficiently combined to produce a representation with higher $I_d$ than any individual layer, then we might expect that new representation to outperform any single layerwise encoding model coming from the same LLM. As linear layerwise encoding models reach their limit, such methods may be necessary to see further benefits.

**Acknowledgements**

This project has received funding from the European Research Council (ERC) under the European Union's Horizon 2020 research and innovation programme (grant agreement No. 101019291) as well as the Dingwall Foundation and a computing gift from the Texas Advanced Computing Center (TACC) at The University of Texas at Austin. This paper reflects the authors' view only, and the funding agency is not responsible for any use that may be made of the information it contains.

The authors would like to thank Alexander Huth, Marco Baroni, Alessandro Laio, and members of the COLT group at Universitat Pompeu Fabra for feedback. The authors additionally thank the organizers of the Brains, Minds, and Machines summer course, where the project was started.

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

Table B.1: Selected GRIDE scales $k$ after performing a scale analysis for intrinsic dimension estimation, for all models and checkpoints tested.

| Model | GRIDE $k$ |
|---|---|
| OPT-125M | 64 |
| OPT-1.3B | 32 |
| OPT-13B | 32 |
| Pythia-6.9B | 16 |
| Pythia ($t =$64000) | 16 |
| Pythia ($t =$32000) | 32 |
| Pythia ($t =$16000) | 32 |
| Pythia ($t =$8000) | 32 |
| Pythia ($t =$4000) | 64 |
| Pythia ($t =$3000) | 64 |
| Pythia ($t =$2000) | 16 |
| Pythia ($t =$1000) | 16 |
| Pythia ($t =$512) | 16 |

# Appendix

## A    Computing resources

Dimensionality and surprisal computation were run on a cluster with 12 nodes with 5 NVIDIA A30 GPUs and 48 CPUs each. Extracting and computing dimensionality on LM representations took a few wall-clock hours per model. Training TunedLens took around 15 minutes per layer, so overall 30 wall-clock hours. We parallelized all computation, and estimate the overall parallelized runtime, including preliminary experiments and failed runs to be around 6 days.

Ridge regression was performed using compute nodes with 128 cores (2 AMD EPYC 7763 64-core processors) and 256GB of RAM. In total, roughly 1,000 node-hours of compute was expended for these models. Feature extraction for language models was performed on specialized GPU nodes similar to the AMD compute nodes but with 3 NVIDIA A100 40GB cards. Feature extraction required roughly 300 node-hours of compute on these GPU nodes. Pycortex [26] and Numpy [27] were used for flatmap visualization and figure generation.

## B    ID Estimation

### B.1    Nonlinear ID

GRIDE operates on ratios $\mu_{i,2k,k} := r_{i,2k}/r_{i,k}$, where $r_{i,j}$ is the Euclidean distance between point $i$ and its $j^{th}$ neighbor. Assuming local uniform density up to the $2k^{th}$ neighbor, the ratios $\mu_{i,2k,k}$ follow a generalized Pareto distribution $f_{\mu_i,2k,k}(\mu) = \frac{I_d(\mu^{I_d}-1)^{k-1}}{B(k,k)\mu^{I_d(2k-1)+1}}$, where $B(\cdot,\cdot)$ is the beta function. The $I_d$ is then recovered by maximizing this likelihood over points $i$ for several candidate scales $k$.

Finally, in order to choose the proper $I_d$, a scale analysis over $k$, which controls the neighborhood size, is necessary: if $k$ is too small, the $I_d$ likely describes local noise, and if $k$ is too large, the curvature of the manifold will produce a faulty estimate. Instead, it is recommended to choose a $k$ for which the $I_d$ is stable [18].

For ID estimation using GRIDE, we reproduce the setup in Cheng et al. [12]. For each model, checkpoint, and layer, we perform a scale analysis. The intrinsic dimension of the manifold is sensitive to the *scale*, or neighborhood size, for which it is estimated [19, 18]. Figure B.1 shows an example, where the GRIDE scale $k$ varies from $2^0$ to $2^{12}$. As recommended in Denti et al. [18], we choose a scale $k$ corresponding in a range where the intrinsic dimension is stable, or plateaus, by visual inspection. For simplicity, we choose one scale $k$ per model. In the particular example in Figure B.1, we choose $k = 2^4$, where the derivative of the curve is closest to 0 for as many layers as possible. Scales chosen for all models are in Table B.1.

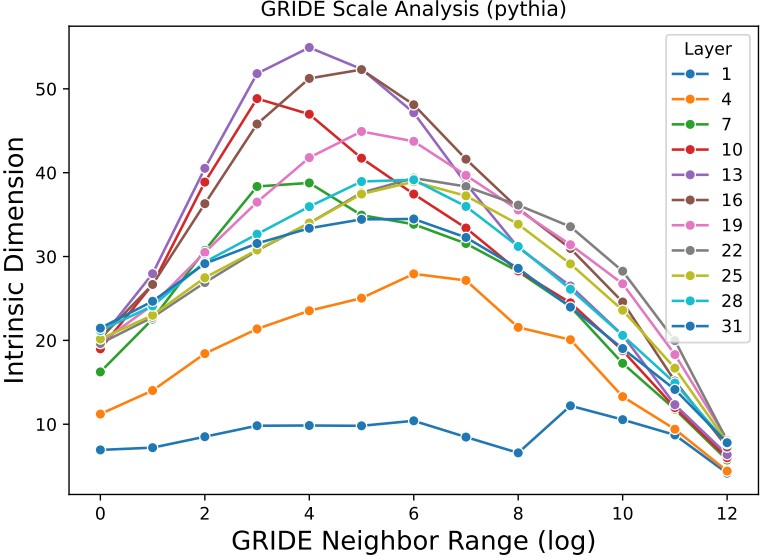

Figure B.1: GRIDE scale analysis for Pythia-6.9B. The estimated intrinsic dimension (y axis) varies according to the chosen scale $k$ (x axis). It is recommended to choose a scale where the local change is minimal, in this case, $k = 2^4$.

## B.2  Linear dimensionality

To compute the linear effective dimensionality of a mean-centered representation matrix $X$, we compute the eigenspectrum $\lambda_1 \geq \lambda_2 \geq \cdots \lambda_D$ of its covariance matrix $X^\top X \in \mathbb{R}^{D \times D}$. Then, the dimensionality given by Principal Component Analysis with a threshold of 0.99 is

$$d_{PCA,0.99}(X) = \min_{i \in 1 \cdots D} \ i \qquad \text{s.t.} \qquad \frac{\sum_{j=1}^{i} \lambda_j}{\sum_{j=1}^{D} \lambda_j} \geq 0.99. \tag{B.1}$$

In words, this is the minimal number of principal components that explain at least 99% of the variance in $X$.

We also compute the Participation Ratio (PR), a non-integer measure of linear dimensionality defined as

$$d_{PR}(X) = (\sum_j \lambda_j)^2 / (\sum_j \lambda_j^2). \tag{B.2}$$

The PR is designed to smoothly interpolate between 1 and $D$: one can verify that when $\lambda_{i \neq 1} = 0$, then $d_{PR}(X) = 1$, and when data are isotropic, that is, $\lambda_i = \lambda_j \ \forall i \neq j$, then $d_{PR}(X) = D$ [21].

## C  Surprisal Estimation

We used the TunedLens [28] implementation by Ghandeharioun et al. [29]. TunedLens ascertains the amount of information linearly encoded in hidden layer $t$ about the next token. To do so, an affine mapping is learned from the last-token hidden representation $h_t$ at layer $t$ as follows:

$$\min_{A_t, b_t} \mathcal{D}_{KL}(f_{>t}(h_t) \ || \ \text{LayerNorm}(A_t h_t + b_t) W_U). \tag{C.3}$$

Here, $A_t \in \mathbb{R}^{D \times D}$, $b_t \in \mathbb{R}^D$ are the learnable parameters of the affine mapping. $W_U$ is the LM's unembedding matrix that maps the final layer to the vocabulary. Finally, $f_{>t}(h_t)$ is the layers of the LM $f$ after layer $t$, producing the model's original distribution over the vocabulary. In the provided code [29], TunedLens is implemented using a direct solver `numpy.linalg.lstsq` on $N = 8000$

randomly sampled sequences from The Pile dataset [16], returning the least squares solution that minimizes the $l_2$-norm between $h_t$ and last layer representation $h_T$. Finally, we compute the next-token surprisals on a validation set of The Pile ($N = 2000$) from the TunedLens-modified hidden layers.

# D  fMRI Methods

MRI data were collected on a 3T Siemens Skyra scanner at The University of Texas at Austin Biomedical Imaging Center using a 64-channel Siemens volume coil. Functional scans were collected using a gradient echo EPI sequence with repetition time (TR) = 2.00 s, echo time (TE) = 30.8 ms, flip angle = 71°, multi-band factor (simultaneous multi-slice) = 2, voxel size = 2.6mm x 2.6mm x 2.6mm (slice thickness = 2.6mm), matrix size = 84x84, and field of view = 220 mm. Anatomical data were collected using a T1-weighted multi-echo MP-RAGE sequence with voxel size = 1mm x 1mm x 1mm.

In addition to motion correction and coregistration [30], low frequency voxel response drift was identified using a 2nd order Savitzky-Golay filter with a 120 second window and then subtracted from the signal. The mean response for each voxel was subtracted and the remaining response was scaled to have unit variance.

# E  LLM Details

Table E.1: LLMs considered for both encoding and dimensionality analysis. All models are causal language models, where each layer consists of a self-attention and an MLP layer. For the Pythia-6.9B model, we considered training checkpoints $t \in \{512, 1000, 2000, 3000, 4000, 8000, 16000, 32000, 64000, 143000\}$.

| LLM | Hidden dimension $D$ | # Parameters | # Layers |
|---|---|---|---|
| OPT | 768 | 125m | 12 |
| | 2048 | 1.3b | 24 |
| | 5120 | 13b | 40 |
| Pythia | 4096 | 6.9b | 32 |

# F   Extended Results

## F.1   Extended Tuned Lens Results

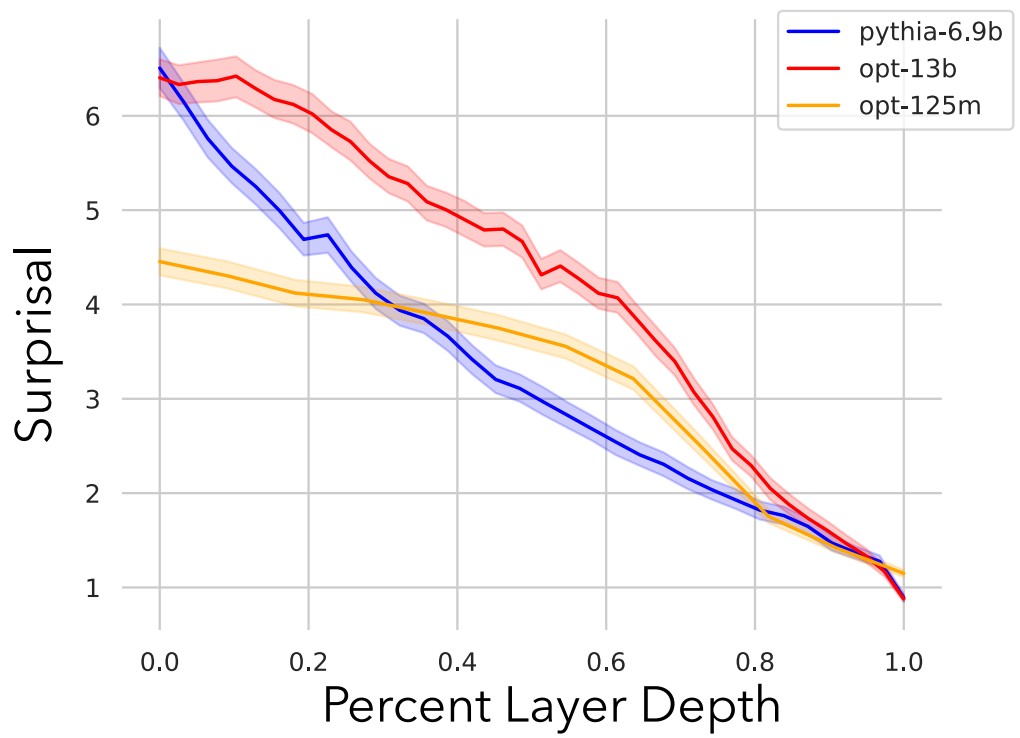

Figure F.1: Remaining tuned lens results for OPT-125, OPT-13B, and Pythia-6.9B

## F.2 Extended Voxelwise ID Correlation Results

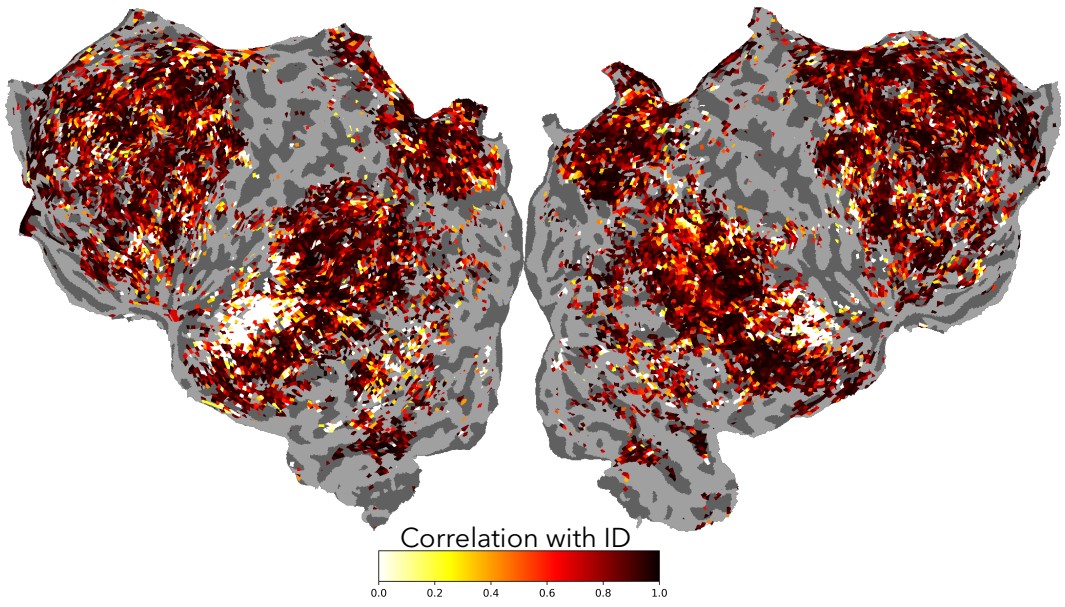

Figure F.2: Voxelwise ID correlation results as in Figure 1c for OPT-125M

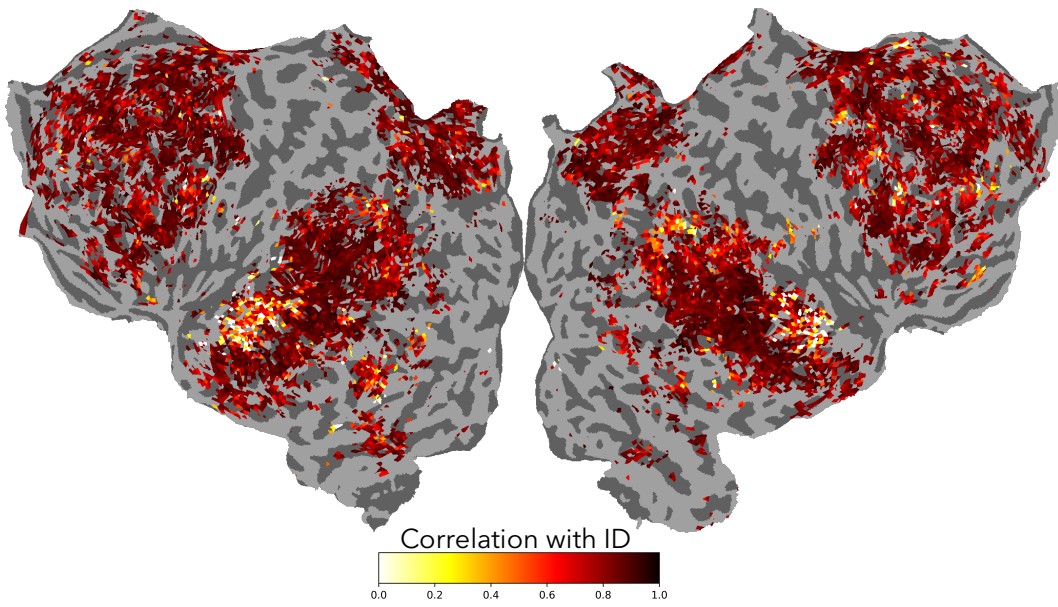

Figure F.3: Voxelwise ID correlation results as in Figure 1c for OPT-13B

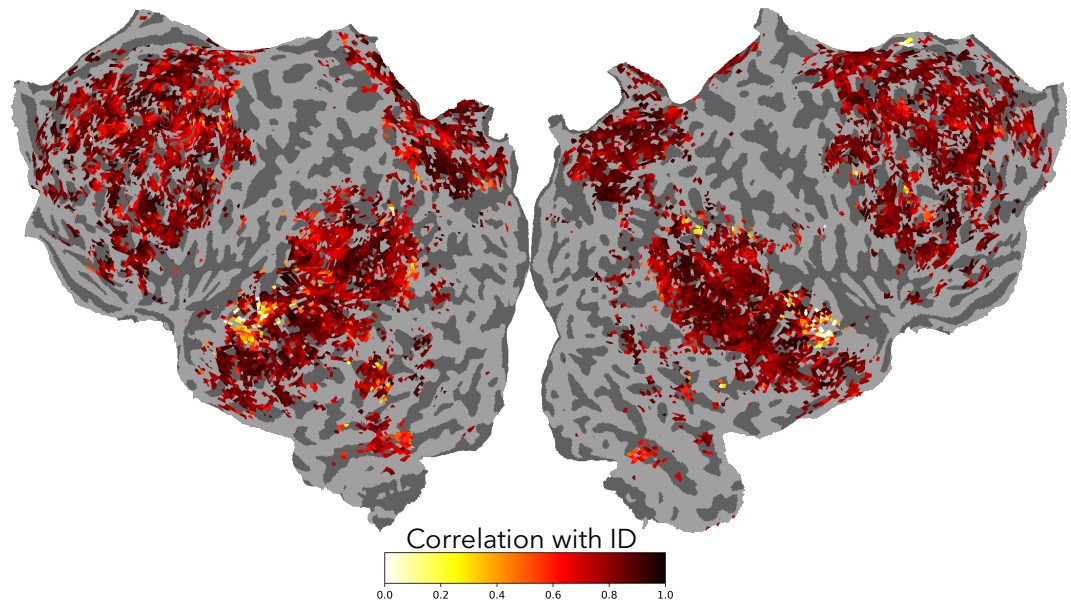

Figure F.4: Voxelwise ID correlation results as in Figure 1c for Pythia-6.9B

### F.3 Extended CKA Results

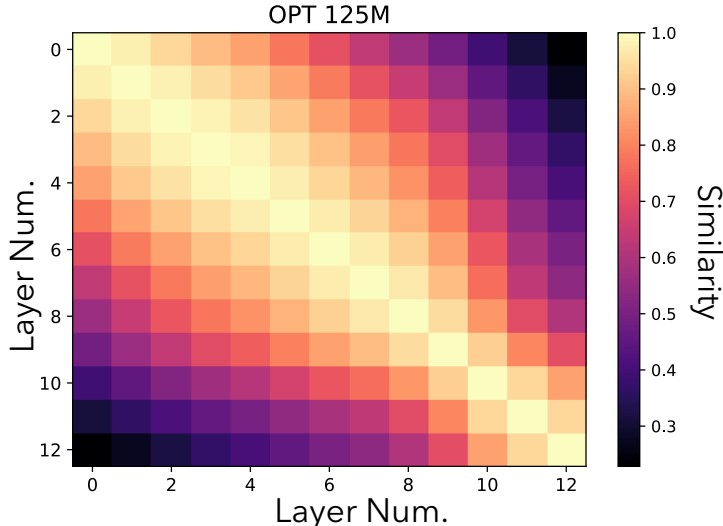

Figure F.5: CKA results as in Figure 1d for OPT-125M

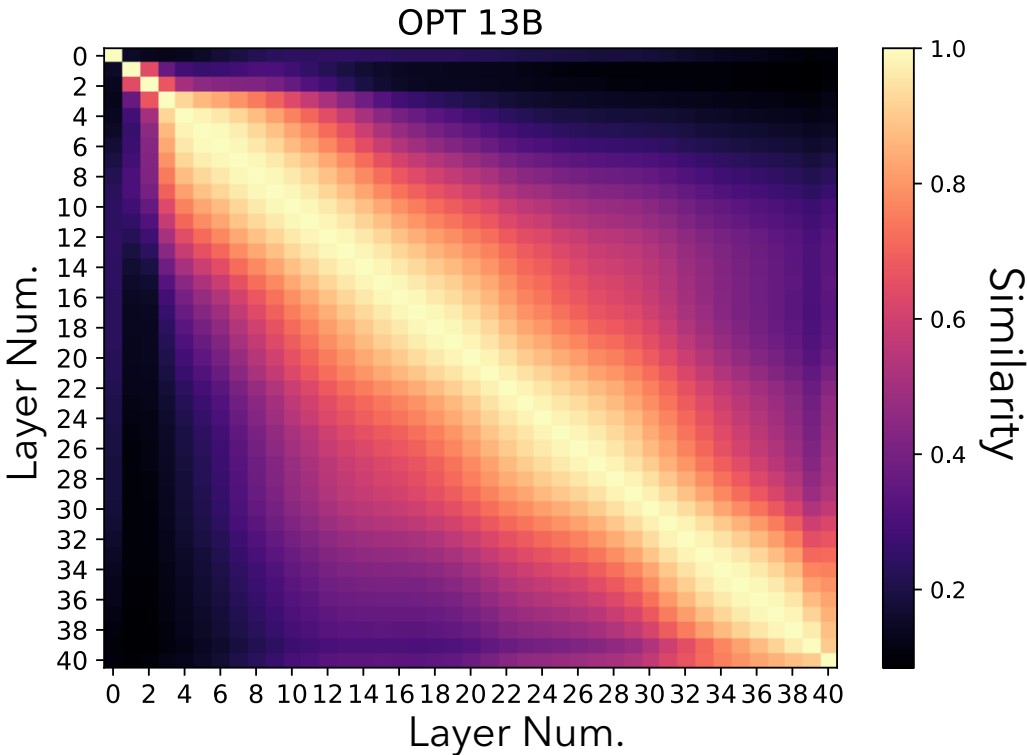

Figure F.6: CKA results as in Figure 1d for OPT-13B

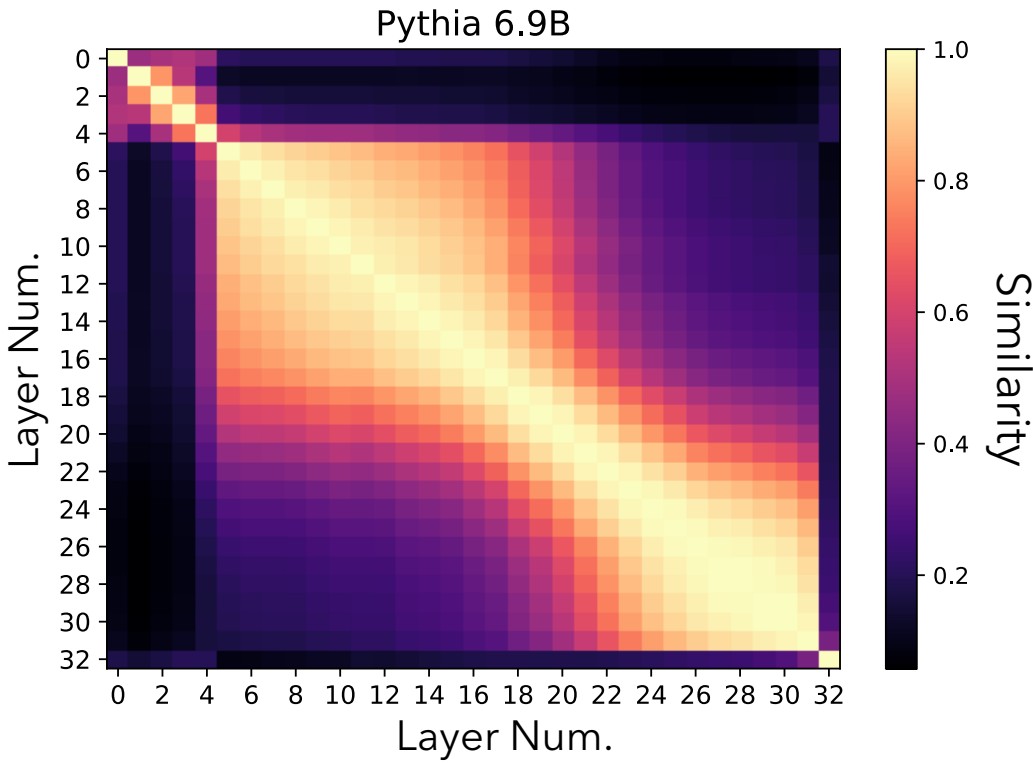

Figure F.7: CKA results as in Figure 1d for Pythia-6.9B

