# OpenReview forum: "Evidence from fMRI Supports a Two-Phase Abstraction Process in Language Models"
_NeurIPS.cc/2024/Workshop/UniReps — UniReps_

### Official Review · Reviewer_cDrW · 2024-09-29
**The topic and results are extremely important and interesting for our understanding of the brain and LLMs.**

**Rating:** 9
**Confidence:** 4

**Review:**

It’s a very interesting study that compares the brain and large language models (LLMs) and explores the underlying reasons for their similar functions. Overall, the author presents a well-structured and engaging paper. I enjoyed reading it and didn’t encounter any major issues. However, I have a few minor suggestions that could help further improve the manuscript.

1. In the neural encoding model training section, you briefly mention [4] for the specific methods used in the encoding model. However, including a brief description of these methods in the appendix or main text would be beneficial, as it would help readers better understand your paper.

2. If Figures 1C, E.2, and E.3 use Pycortex for visualization, please cite the relevant paper.

3.  Could you add a table or brief descriptions for each LLM in the appendix if there isn’t enough space in the main text? This would help readers quickly grasp the key information about each model and enhance the clarity of your paper.

4. Please add an "Appendix Section" text label on page 7, as it appears to be missing.

---

### Official Review · Reviewer_To4W · 2024-10-06
**A study of why certain layers are more aligned with the brain than others, revealing the distinct roles that certain layers play and the effects that that role has in shaping dimensionality and alignment.**

**Rating:** 9
**Confidence:** 3

**Review:**

Synopsis:

This paper studies the phenomena of brain-LLM similarity, but goes beyond the 'what' (similarity measurement) to an investigation of the 'why'. The authors look into different reasons why middle layers may be better aligned with the brain, finding that earlier layers function as an abstraction process and later layers function as an extraction process for prediction (roughly speaking). The layers that work to develop better abstractions of the input are well aligned. The authors back this up with strong methodology and thorough experiments.


Pros:
- I think that the methodology is very strong. Measuring between different models (OPT and Pythia) is good, and the use of the different checkpoints of Pythia to see how learning effects intrinsic dimensionality and brain-model alignment is very interesting. I think this could nicely lead into some future work if you were to also study grokking, etc.

- Tying this into some of the LLM interpretability literature is a good touch. I have seen some studies describing different phases / phase changes in models as they learn, and I was pleased to see that your results provide further evidence of this.

- I also think that this paper is quite well written, with very good graphics. The contextualization of the problem is also particularly well done.

- It seems interesting that next-token prediction seems to explain this less than the intrinsic dimensionality. Perhaps a good direction for future work would be to explore how different training objectives alter the intrinsic dimensionality through time. This may require quite a lot more experiments and compute though - I'm not sure if models with provided checkpoints like Pythia are abundant for studying this yet.


Cons:

- Training on some more models would have been nice, especially with different training objectives or input modalities. However, given the compute already expended for these results, this is not a major objection.

Overall:

Overall I'm really quite pleased with your work. I think it's easily strong enough to have gone in the proceedings track instead of the extended abstracts track, but perhaps you want to extend on it and publish a lengthier version elsewhere. I'm quite happy with your decision to dig deeper into why certain layers may be better aligned, and I haven't seen much work (if any) in this direction yet. Keep up the nice work!

---

### Official Review · Reviewer_qucJ · 2024-10-07
**new approaches to understanding the stages of LLM inference**

**Rating:** 8
**Confidence:** 4

**Review:**

Paper explores the relationship between the representations learned by large language models (LLMs) and human brain activity as measured by fMRI by proposing both follow a two phase process of abstraction and prediction. Though the idea that LLMs process in multiple stages is not new (eg: Lad 24 = https://arxiv.org/abs/2406.19384) - this paper puts this work into new perspective in several ways. The first is by looking this phenomenon with two different metrics - intrinsic dimensionality and surprisal; I found both of these interesting ways of framing and measuring the workings of the inner layers. The second is by comparing both to fMRI data to make an argument that this transition point of highest intrinsic dimensionality is the area with the most brain similarity. I also found the study on how these stages develop during the course of training to be strong support for their framework.

I am not well versed in interpreting fMRI data and not convinced that these two phases of abstraction and prediction are universal. But the authors make an interesting case for this and I otherwise found this paper clear and on topic for the workshop as well as densely packed with results in the context of a four page extended abstract.

---

### Official Review · Reviewer_57fd · 2024-10-07
**This study aims to connect recent ideas from interpretability literature on language model abstraction with how the brain might be processing language (using fMRI data), tying the notion of dimensionality of activations to explain recent empirical observations that have yet to be explained.**

**Rating:** 7
**Confidence:** 3

**Review:**

**Quality**:
The study appears to be well-executed with rigorous methodology. The authors use state-of-the-art language models, appropriate dimensionality estimation techniques for the activations, and established fMRI encoding methods. The analysis is thorough and considers multiple models, metrics, and controls.

**Clarity**:
The paper is generally well-written and logically structured. The methods and results are explained clearly, with helpful figures to illustrate key findings. Some technical details are appropriately relegated to appendices, including details related to compute requirements, dimensionality formulas, etc. However, a few sections (e.g., parts of the discussion) could benefit from minor clarification or expansion. For instance, the low correlation in Table for PR-d and the encoding performance is not explained later in the results section.

**Originality**:
The work presents several novel contributions to my knowledge:

* Linking intrinsic dimensionality of language model layers to their ability to predict brain activity (correlate with encoding model performance)
* Providing evidence for a two-phase abstraction process in language models using fMRI data, via the notion of dimensionality
* Demonstrating what this reveals about the abstraction and prediction process in LLMs

These insights offer a new perspective on why intermediate layers of language models are often best at predicting brain activity, along with what the higher-level language areas might be doing in the brain.

**Significance**:

This research has potentially important implications for both neuroscience and AI, and the particular topic of this workshop:

* It provides new evidence about how the brain processes language, suggesting similarities with the abstraction processes in language models.
* It offers insights into the internal representations of language models that could inform future model development and interpretation.

**Pros**:

1. Novel integration of intrinsic dimensionality analysis with fMRI encoding models
2. Comprehensive analysis across multiple language models and model sizes (from 125m to 13b)
3. Examination of how representational properties emerge during training via layer-wise analysis
4. Strong empirical support for the main hypotheses using multiple metrics to build robustness, including evidence to suggest the compositional phase contributing to increased encoding performance (while prediction phase deteriorates performance)
5. Many hours of fMRI recordings being studied across subjects, over diverse sets of stories

**Cons**:
1. fMRI data from only 3 subjects, which may limit generalizability of the results
2. Focuses mainly on correlational relationships between dimensionality and encoding model performance; whether this is the mechanism behind alignment is still unclear (Figure 2 particularly appears to suggest some uncertainty on this from later layers)
3. Some aspects of the discussion could be expanded, particularly regarding alternative interpretations of what the dimensionality indicates about the representations in the brain.

In conclusion, this paper presents a high-quality, original study that would be a great fit to the UniReps workshop to scale our understanding and alignment of language processing in both artificial and biological systems. While there are some limitations, the strengths of the work outweigh its weaknesses, and it opens up promising avenues for studying the mechanisms behind the trends in LLM-brain alignment in recent years, in addition to providing support for new hypotheses on how the brain processes language.

---

### Decision · Program_Chairs · 2024-10-10

**Decision:**

Accept (Oral)

**Comment:**

In light of the positive reviewers' feedback and relevancy of the submission, we are pleased to accept this paper for presentation at UniReps 2024. We kindly ask the authors to incorporate the reviewers' suggestions and feedback in the final camera-ready version of the manuscript.